# Epitope-targeting platform for broadly protective influenza vaccines

**David F. Zeigler[¤], Emily Gage[¤], Christopher H. Clegg[iD] ***

TRIA Bioscience Corp., Seattle, Washington, United States of America

¤ Current address: Lonza Pharma and Biotech, Bend, Oregon, United States of America
* cclegg@triabio.com

**Data Availability Statement:** All relevant data are within the paper and its Supporting information files.

**Funding:** This research was supported in its entirety by the Center of Disease Control (www.cdc.gov) under award number R43IP001108 (CC).

## Abstract

Seasonal influenza vaccines are often ineffective because they elicit strain-specific antibody responses to mutation-prone sites on the hemagglutinin (HA) head. Vaccines that provide long-lasting immunity to conserved epitopes are needed. Recently, we reported a nanoparticle-based vaccine platform produced by solid-phase peptide synthesis (SPPS) for targeting linear and helical protein-based epitopes. Here, we illustrate its potential for building broadly protective influenza vaccines. Targeting known epitopes in the HA stem, neuraminidase (NA) active site, and M2 ectodomain (M2e) conferred 50–75% survival against $5LD_{50}$ influenza B and H1N1 challenge; combining stem and M2e antigens increased survival to 90%. Additionally, protein sequence and structural information were employed in tandem to identify alternative epitopes that stimulate greater protection; we report three novel HA and NA sites that are highly conserved in type B viruses. One new target in the HA stem stimulated 100% survival, highlighting the value of this simple epitope discovery strategy. A candidate influenza B vaccine targeting two adjacent HA stem sites led to >$10^4$-fold reduction in pulmonary viral load. These studies describe a compelling platform for building vaccines that target conserved influenza epitopes.

## Introduction

The rapid mutation rate of influenza viruses fuels seasonal epidemics that cause >0.25 million deaths annually and facilitates occasional pandemic outbreaks that can lead to >20 million fatalities [1–5]. Human infections are caused by type A (IAV) and B (IBV) viruses. Strains are classified based on antigenic variation in hemagglutinin (HA) and neuraminidase (NA), the two major surface proteins which respectively enable viral fusion and budding. While vaccination is the best prophylactic, there is still tremendous need for improvement. The fundamental problem with current vaccines is that they elicit antibodies to mutable regions of the HA head that have limited homology between strains [6, 7]. Thus, antigenic mismatch between vaccine and circulating strains severely limits effectiveness. Moreover, selective pressure on unstable epitopes favors escape mutants with substitutions that abrogate antibody binding, thereby undermining long-term protection. This continual antigenic drift forces vaccines to be updated annually based on prediction of the strains that will dominate the upcoming year [8].

The funders had no role in the study design, data collection and analysis, decision to publish, or preparation of the manuscript. The authors are employed by TRIA Bioscience Corp, which provided support in the form of salaries but did not have any additional role in the study design, data collection and analysis, decision to publish, or preparation of the manuscript. The specific roles of these authors are articulated in the 'author contributions' section. There was no additional external funding received for this study.

**Competing interests:** All authors are employees of TRIA Bioscience Corp. DZ and CC are co-inventors on a pending patent (Applicant: TRIA Bioscience Corp., "Synthetic carrier compositions for peptide vaccines", WO/2020/047107) pertaining to peptide composition of matter and methods of use. This does not alter our adherence to PLOS ONE policies on sharing data and materials.

Influenza vaccines that can stimulate long-term broadly protective humoral immunity are needed.

Foundational studies have identified sequences on the three surface proteins (HA, NA, and M2, a proton channel critical for viral replication) that are highly conserved across strains, making them potential targets for broadly protective vaccines. These sites do not naturally stimulate appreciable antibody responses and much effort has been devoted to overcoming this problem [9]. One of the most common targets is the HA stem, which is substantially more conserved than the head subunit [10]. Many antigens have been designed to focus antibody responses on this locale. For instance, hyper-glycosylated, computationally optimized (COBRA), and mosaic HA vaccines are in preclinical development [11–15]. A phase I trial testing headless stem-ferritin nanoparticles is underway (clinicaltrials.gov, NCT03814720) [16]. Interim phase I results for chimeric HA-based vaccines showed suboptimal memory responses to the stem [17]. Epitope-targeting platforms–e.g., conjugate, virus-like particle (VLP), and peptide vaccines–have also been tested in clinical trials [6, 7, 18–20]. These antigens present conserved epitopes outside their native context, which avoids interference from the HA head and other mutation-prone domains [21]. Epitope-targeting platforms have largely concentrated on M2e, the exposed ectodomain of M2 that is conserved separately within IAV and IBV [20, 22–24]. These vaccines elicit M2e-specific antibodies in humans, but at inefficacious levels [25]. Furthermore, it is unclear whether targeting M2e alone can stimulate sufficient protection [6, 26–31]. Designing antigens that elicit strong focused responses to conserved influenza epitopes remains a major challenge.

We are developing a vaccine platform based on peptide nanoparticles that induce functional antibodies to small molecules and protein-based epitopes. This technology is based on peptide monomers (~70 amino acid) made using solid phase synthesis that consist of three functional domains: an amphipathic helix that drives nanoparticle self-assembly, two universal CD4 T cell epitopes that mediate high-affinity and long-lived antibody responses, and a targeted B cell epitope at one or more predetermined sites [32–35]. To enhance immunogenicity, the nanoparticles are paired with GLA-SE, an adjuvant consisting of a toll-like receptor-4 agonist in a stable emulsion [36]. This adjuvant promotes $T_H1$-mediated antibody class switching and antibody-dependent cellular cytotoxicity (ADCC), which are requirements for protection mediated by non-neutralizing antibodies to sites such as M2e [20, 37]. Previously, we used the platform to target Helix A, a conserved site on the HA stem that is bound by broadly neutralizing antibodies [35]. The vaccine partially protected mice from a lethal H1N1 challenge, confirming the antiviral potential of the platform. This proof-of-concept also illustrated the platform's unique ability to present helical epitopes in their native conformation, which is difficult for most epitope-targeting platforms [38, 39]. Herein, we demonstrate how this platform can be used to construct pan-subtype influenza vaccines.

## Materials and methods

### Ethics

This study was carried out in strict accordance with the recommendations in the Guide for the Care and Use of Laboratory Animals of the National Institutes of Health, the US Public Health Service (PHS), and the Association for Assessment and Accreditation of Laboratory Animal Care International (AAALAC). Protocol #2019–17 was approved by the Institutional Animal Care and Use Committees (IACUC) of the Infectious Disease Research Institute which operates under a currently approved Assurance #A4337-01, which is in accordance with PHS Policy for Humane Care and Use of Laboratory Animals.

## Peptides

Peptides were synthesized by Bio-Synthesis Inc. (Lewiston, TX). All peptides contained N-terminal acetyl units and chloride counterions. The peptide monomer used in these experiments contains 4 IKKIEKR heptad repeats fused to TCEs selected from Measles virus F2 protein (LSEIKGVIVHRLEGV) and Hepatitis B surface antigens (FFLLTRILTIPQSLD) [40, 41]. These peptides were made using standard SPPS chemistry with Fmoc protecting groups. The M2e 1xC-terminus antigen was made by synthesizing the M2e sequence (SLLTEVETPT) onto the *C*-terminal of the coiled-coil domain with a Gly linker. Peptides with two epitope copies on the self-assembly domain were synthesized in the following manner: 1), the target B cell epitope sequence was synthesized separately and purified with its reactive functionalities protected; 2), the self-assembling peptide monomer was synthesized with labile protecting groups on the desired Lys sidechains (heptad *f* positions) located in the first and fourth heptad repeats, 3), while still on the SPPS solid support, these Lys ζ-amine groups were deprotected and bonded to the *C*-terminal carboxylic acid of the B cell epitope sequence using standard SPPS amide formation chemistry, and 4), all remaining protecting groups were removed to yield the final peptide. This strategy was used to build antigens targeting $M2e_{IAV}$, $M2e_{IBV}$, $NA_{222}$, $NA_{238}$, $HA1_{27}$, and $HA1_{231}$. The Helix $A_{H1}$ and Helix $A_{IBV}$ sequences were synthesized onto the N-terminus of the carrier peptide during SPPS.

## Dynamic light scattering

DLS spectroscopy was performed using a Zetasizer Nano (Malvern Instruments, UK) with a 4 mW He–Ne laser (633 nm) and a fixed detection angle (173˚). To avoid interference from the adjuvant emulsion, peptides were formulated without GLA-SE in PBS or MOPS (100 mM, 50 mM NaCl, pH 7.5) at the concentration used for immunizations. Solutions were filtered through a 0.2 μm nylon membrane and loaded into a plastic microcuvette. Measurements were carried out in general purpose model with the following parameters: material setting was protein (refractive index = 1.440), dispersant setting was water (viscosity = 0.8872 cP, refractive index = 1.330), 10 cycles averaged per measurement, and 30 second temperature equilibration at 25˚C.

## B cell epitope discovery

Protein sequences of influenza B viruses were created using human isolate sequences from the NIAID Influenza Research Database and Global Initiative on Sharing All Influenza Data [42, 43]. Sequences were sorted to exclude duplicate sequences. The final data set contained 3182 HA and 3331 NA sequences. Epitopes were aligned with this library using MUSCLE to identify contiguous regions of homology between strains [44]. Homologous regions were identified on published X-ray diffraction structures of representative HA (B/Yamanashi/166/1998) and NA (B/Brisbane/60/2008) proteins. The numbering of $NA_{238}$, $HA1_{27}$, and $HA1_{231}$ are relative to the position of the start codon Met residue in the respective proteins.

## Sequence homology

Homologous and nonhomologous substitutions were tallied for each residue in the putative B cell epitopes. Amino acid identity and frequency at each position was calculated. The relative prevalence of each substitution was used to generate a visual representation of conservation.

## Protein modeling

Files depicting X-ray crystallography structures of representative HA [42] and NA [43] proteins were downloaded from the RCSB Protein Data Bank (PDB). These PDB files were

opened with the Visual Molecular Dynamics (VMD) viewer [45]. Proteins were depicted with the ColorID coloring method and Surf drawing method. Glycosylation was depicted using bonds.

## Animals

Mice (Charles River Laboratories) were housed and handled by highly trained researchers under specific pathogen-free conditions with easy access to food and water within the Infectious Disease Research Institute vivarium (Seattle, WA). Since the vaccine's CD4 T cell epitopes bind promiscuously to a broad repertoire of MHCII molecules, our experiments employed outbred female CD-1 mice (6–8 wks) to more accurately model immune responses in a genetically diverse population like humans. Peptides were dissolved phosphate-buffered saline (PBS) or MOPS (100 mM, 50 mM NaCl, pH 7.5) buffers and filtered through a 0.2 µm nylon membrane to create immunization stocks. Final concentrations of these stocks were determined by amino acid analysis (AAA Service Laboratory, Damascus, OR). The peptides were combined on the day of immunization with GLA-SE adjuvant containing 5 µg of the synthetic TLR4 agonist, GLA, formulated in a final 2% oil-in-water stable emulsion. The adjuvant was provided by Immune Design Corp (Seattle, WA). Mice, which were inoculated under isoflurane anesthesia, received 10 µg of each indicated peptide diluted in 100 µL total volume, 50 µL of which was injected in each hind limb using a prime-boost regimen (d0 and d21). Serum was collected on d35 and used to measure antibody responses. Influenza challenge experiments were performed by infecting mice intranasally with $5LD_{50}$ dose of A/California/07/2009 or B/Florida/04/2006 in 50 µL PBS. Mice were monitored daily for 14 days to measure overall health, body weight changes and survival rates. According to the humane endpoint guideline, mice losing 25% of their body weight relative to the baseline weight were euthanized immediately by carbon dioxide overdose followed by cervical dislocation (euthanized mice, n = 262; found dead mice, n = 26). Mice were monitored for weight loss and other signs of virus induced morbidity daily and sacrificed if weight loss exceeded 25% of initial body weight. Monovalent challenge data represents three combined experiments with $n_{total}$ = 18 ($M2e_{IAV}$), 25 ($NA_{222}$), 22 (Helix $A_{H1}$), 21 ($M2e_{IBV}$), 16 ($NA_{328}$), 21 (Helix $A_{IBV}$), 20 ($HA1_{27}$), and 20 ($HA1_{231}$). The bivalent studies were single experiments (n = 10 mice/group for M2e + Helix A bivalents and n = 11/group for $HA1_{27}$ formulations). Each mouse received 10 µg of the indicated peptides. $HA1_{27}$ monovalent and $M2e_{IBV}$ + Helix $A_{IBV}$ bivalent groups (n = 3/group) were included as pulmonary controls for the second bivalent study. For the pulmonary analysis, whole lungs of a subset of mice (n = 3/group, chosen randomly from groups receiving $HA1_{27}$ formulations) were flash frozen on day 4 post infection for viral titer determination. Briefly, frozen lungs were homogenized using the gentleMACS™ Dissociator M tubes in 1mL sterile PBS and viral titers determined by 50% tissue culture infectious dose in Madin-Darby canine kidney (MDCK) cells.

## Antibody assays

Serum samples were serially diluted 5-fold from 1/20 in blocking buffer (3% BSA in PBST) and IgG endpoint titers were assayed by ELISA using previously-reported methodologies [32]. Endpoint titers were calculated using GraphPad Prism (GraphPad Software, San Diego, CA). Antibodies to $M2e_{IAV}$, $M2e_{IBV}$, $NA_{220}$, $NA_{238}$, $HA1_{27}$, and $HA1_{231}$ were detected using cysteine-terminated synthetic peptides conjugated to BSA through maleimide crosslinking chemistry. Helix A titers, as well as cross-reactivity of $HA1_{27}$ and $HA1_{231}$ antisera, were measured using recombinant HA from A/California/07/2009 or B/Malaysia/2506/2004 (Protein

Sciences). $NA_{238}$ antisera was screened against an available recombinant NA (A/Thailand/1 (KAN-1)/2004) with a sequence nearly identical (PRPNDGT) to the $NA_{238}$ epitope.

## Plaque reduction neutralization titer (PRNT)

Serum samples from immunized mice were inactivated by incubation at 56 ˚C for 30 min. Inactivated serum samples were serially diluted two-fold in DMEM medium without FBS in a 96-well beginning with a 1:2 dilution in a total volume of 100 μL. Following serum dilution, 100 μL of diluted B/Florida/4/2006 virus (50 pfu) was added to all serum samples with TPCK-trypsin (1 μg/mL). Virus: serum mixtures were incubated at 37 ˚C for 60 min. Following incubation, virus−serum mixtures were incubated with MDCK cell monolayers (200 μL/well) in 6-well plates at 33˚C for 60 min with rocking to distribute the medium every 15 min. Wells were overlaid with 1% agarose-MEM and incubated for 3 days at 33˚C in a $CO_2$ incubator. Following this incubation, plaques were fixed with 4% paraformaldehyde (PFA) and stained with crystal violet prior enumeration. Negative (media only) and naive murine serum samples were also assessed. Neutralizing antibody titers are presented as the highest total serum dilution capable of reducing the number of plaques by 50% compared to a virus only control ($PRNT_{50}$).

## NA-Star assay

Serum samples from $NA_1$ and $NA_2$ immunized and naive mice were assayed for NA enzymatic inhibition using the NA-*Star* influenza neuraminidase inhibitor resistance detection kit (Applied Biosystems). To measure sera-mediated inhibition, immunized and naive sera was serially diluted two-fold in NA-Star assay buffer in white, flat-bottom, 96-well cell culture plates. Virus (B/Florida/04/06 or B/Malaysia/2506/04) was diluted to the determined $3EC_{50}$ (half-maximum effective concentration) and 25 μL was added to each well. The plates were incubated for 30 min at 37 ˚C. Data points were expressed as percent inhibition of maximal NA enzymatic activity, which was determined by the activity of virus without the addition of sera. ELISA signals were fit with an inhibition regression algorithm and $IC_{50}$ values determined using GraphPad Prism.

# Results

## Maximizing immunogenicity to an M2e-targeting antigen

We have previously reported that peptides synthesized with one B cell epitope located at either the N- or C-terminus induced equivalent antibody responses [35]. To test whether immunogenicity could be improved by increasing B cell epitope multiplicity [32, 34, 46, 47], mice received a prime boost immunization (S1 Fig) with peptides containing either a c-terminal $M2e_{IAV}$ epitope or two $M2e_{IAV}$ epitopes located on the self-assembly domain (S1 Table). As indicated in S2 Fig, antibody titers were $>10^3$-fold higher in animals receiving peptides containing two $M2e_{IAV}$ epitopes. This result provides further evidence that increasing epitope valency enhances B cell receptor engagement and resultant immune responses.

## Monovalent and bivalent vaccines targeting known influenza epitopes

We next investigated whether the 2x-self-assembly domain template could generate protective responses to other linear influenza epitopes, including $M2e_{IAV}$ and $M2e_{IBV}$ [20, 48–51] and a sequence lining the NA active site, $NA_{222}$ (S2 Table), that is nearly 100% conserved across all influenza subtypes [52]. Also included in this study were two stem-targeting peptides (the previously reported Helix $A_{H1}$ and a new pan-IBV antigen, Helix $A_{IBV}$) [53–55]. The Helix A

monomer designs contain one B cell epitope copy at the *N*-terminus. This design utilizes the natural helicity of the peptide's self-assembly domain to constrain and present the epitope sequence as a helix [35]. Dynamic light scattering (DLS) verified each peptide formed nano-particles (20–40 nm mean hydrodynamic diameters) in aqueous buffer (S3 Fig). We have previously verified that peptides lacking the self-assembly domain fail to reach these size distributions (data not shown). Antibody titers induced by each epitope were comparable (Fig 1A and 1B), although Helix $A_{IBV}$ yielded more variable titers to recombinant HA than Helix $A_{H1}$. Mice were then challenged with H1N1/A/California/07/2009 (Fig 1C and 1D) or B/Florida/04/2006 (Fig 1E and 1F). Respectively, M2e, Helix A and $NA_{222}$ vaccines conferred approximately 75%, 70% and 50% survival regardless of the challenge strain, although weight loss trends across experiments were indistinct. This consistency exemplifies the versatile "plug-and-play" nature of the platform and substantiates its potential for building antiviral vaccines.

The partial protection conferred by these vaccines signaled that combining peptide antigens may further improve efficacy. To test this concept, the two best antigens (M2e and Helix A) were mixed to create pan-H1 and -IBV formulations. Peptide mixtures exhibited ~30 nm diameters (S4 Fig) by DLS, suggesting co-formulation does not interfere with assembly or cause aggregation. Bivalent formulations stimulated antibodies to each epitope (Fig 2A and 2B), boosted survival to 90% (Fig 2C and 2E) and statistically decreased weight loss (Fig 2D and 2F) over control mice 1–2 days earlier than their composite monovalent vaccines. Dose-ranging studies comparing 20 μg monovalent formulations to the bivalent made from 10 μg of each peptide have confirmed that the improved protection conferred by bivalent formulations is not due to peptide dose (data not shown). These data suggest that targeting two influenza epitopes simultaneously has the potential to improve protection.

## Identification and validation of conserved influenza B antibody epitopes

These encouraging results suggested that bivalent vaccine efficacy might be improved by substituting more protective antigens in the formulation. To identify potential new epitopes, homologous amino acid stretches within IBV HA and NA were located on published X-ray diffraction structures [56, 57]. Suitable antibody targets were ≥6 amino acids in length, surface-exposed, had a linear or looped conformation, and did not possess a glycosylation motif (Asn-X-Ser/Thr/Cys). Three sites ($HA1_{27}$, $HA1_{231}$, $NA_{328}$) were identified that showed strong sequence homology across >3000 IBV strains (S5 Fig). $HA1_{27}$ (Fig 3A and 3C) is situated along a raised ridge on the HA stem and lies end-to-end with the Helix A epitope, which is rotated toward a recessed hydrophobic pocket. $HA1_{27}$ abuts conserved glycosylated Asn residues (N25, N301, N330). $HA1_{231}$ (Fig 3A and 3B) is a loop flush with the HA head. It is adjacent to several potential glycosylation sites that vary by strain (e.g. N59, N145, N163). Each residue in these HA epitopes is >99% conserved across type B viruses. The $NA_{328}$ epitope (Fig 3D and 3E) lies near but is more surface exposed than $NA_{222}$. To gauge their antiviral activity, peptides targeting these 3 putative epitopes were vetted in vivo. Two copies of each sequence were grafted onto identical locations within the peptide monomer (S3 Table) and prior to immunization, their ability to form nanoparticles was confirmed (S6 Fig), as was their sequence conservation with the virus challenge strain (B/Florida). Importantly, these peptides induced epitope-specific antibodies that bound recombinant protein (S7 Fig) and protected against virus (Fig 3F and 3G). As indicated, $HA1_{27}$ conferred 100% survival, while $HA1_{231}$ and $NA_{328}$ groups exhibited 80% and 65% survival, respectively. Average weight loss in the $HA1_{27}$ and $HA1_{231}$ groups remained less than 10%, exhibiting statistically better protection than the $NA_{328}$ vaccine. These results demonstrate a general ability to identify and target novel antibody epitopes using protein sequence and structural data.

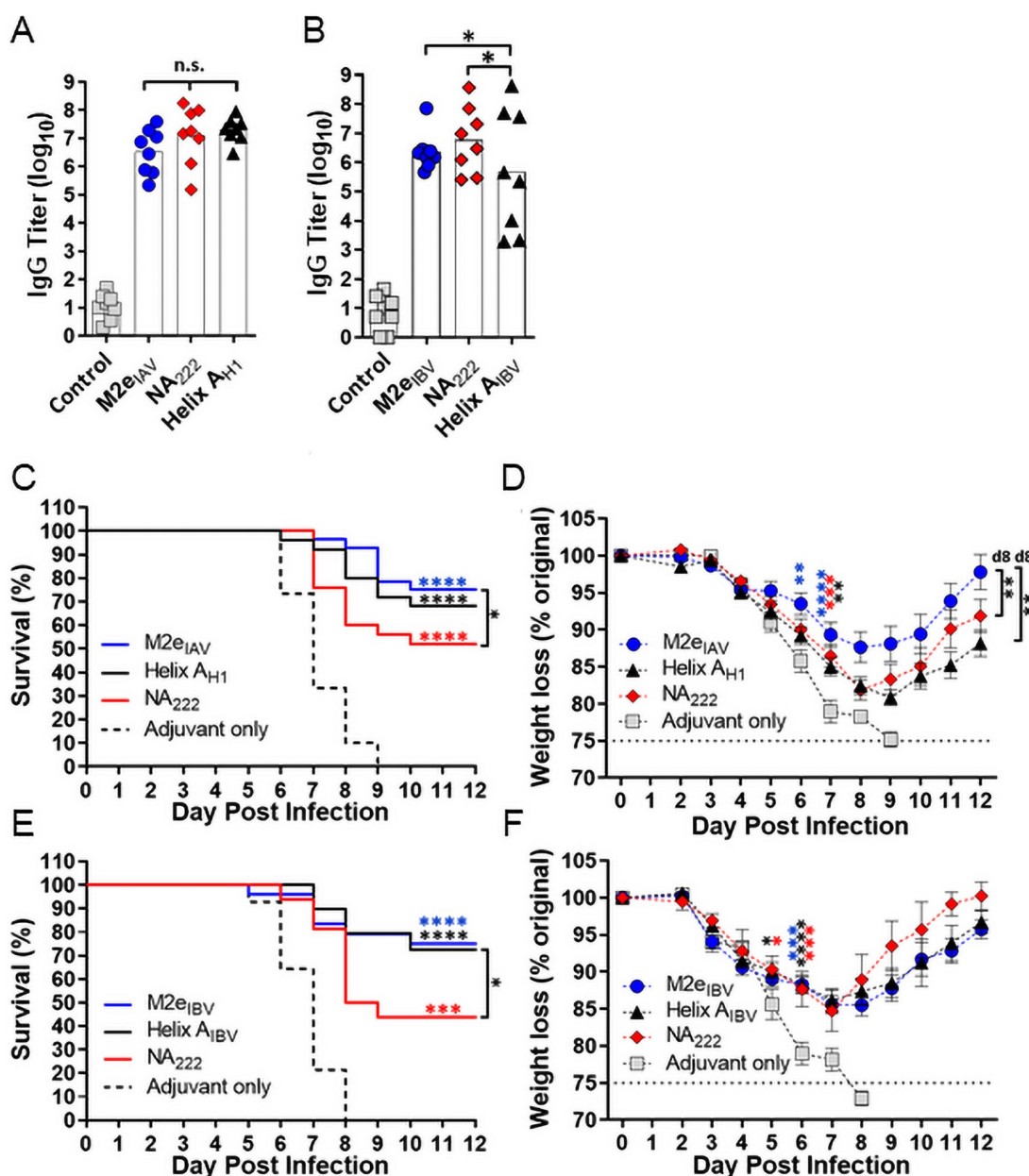

**Fig 1. Peptides targeting conserved IAV and IBV epitopes stimulate robust antibody responses and confer protection against lethal challenge.** CD-1 mice (n = 8) were immunized in a prime-boost regimen with the indicated (*A*) IAV or (*B*) IBV peptides plus GLA-SE (or GLA-SE only as a control) and d35 sera was assayed for titers by ELISA. A one-way analysis of variance (ANOVA) followed by Tukey's multiple comparisons test was used for statistical analysis (*$P<0.05$, n.s. not significant). On d42, mice were challenged with (*C,D*) A/California/07/2009 or (*E,F*) B/Florida/04/2006 and then monitored for survival and weight loss plotted as mean ± S.E.M. Monovalent challenge data was compiled from three experiments (n = 5-8/experiment). Survival curves were compared by log-rank Mantel-Cox test. Data from each weight loss time point were compared by one-way ANOVA followed by Dunnett's multiple comparisons test. Color coded asterisks without brackets denote significance between control and indicated test group. Brackets indicate comparison between test groups. For weight loss, significance over the control is shown until maximum difference and comparison between test groups was maximum on the designated day (*$P<0.05$, **$P<0.01$, ***$P<0.001$, ****$P<0.0001$).

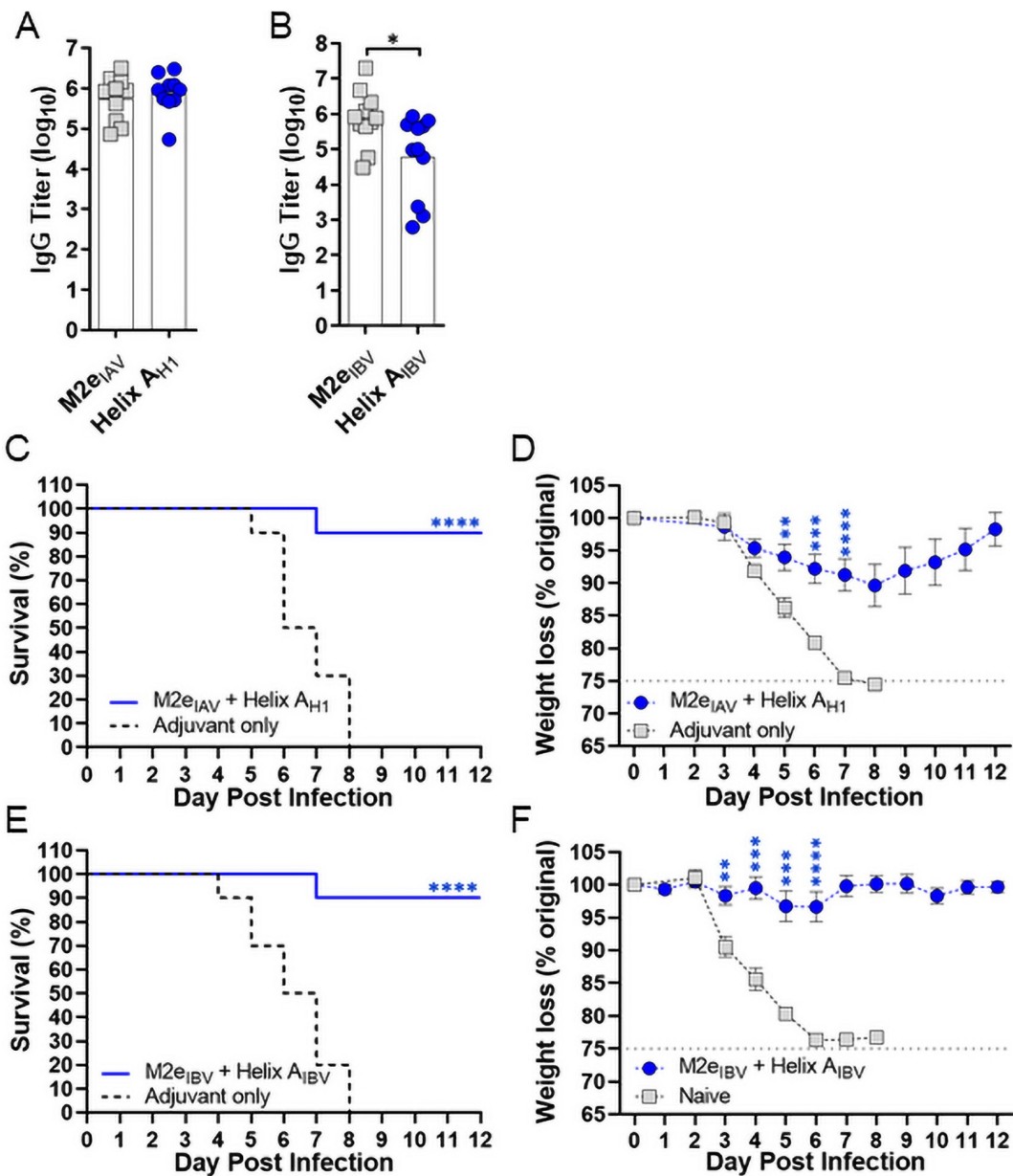

**Fig 2. M2e + Helix A coformulations stimulate epitope-specific antibody responses to both targets and improve antiviral protection.** CD-1 mice (n = 11) were immunized with (*A*) M2e$_{IAV}$ + Helix A$_{H1}$ or (*B*) M2e$_{IBV}$ + Helix A$_{IBV}$ formulations (10 μg/peptide) in GLA-SE and d35 sera was assayed for antibody titers by ELISA. Statistical differences were calculated with an unpaired two-tailed t-test (*P<0.05). On d42, mice were challenged with (*C,D*) A/California/07/2009 or (*E,F*) B/Florida/04/2006 and then monitored for survival and weight loss plotted as mean ± S.E.M. Survival curves were compared by log-rank Mantel-Cox test. Data from each weight loss time point were compared by one-way ANOVA followed by unpaired two-tailed t-test. Asterisks denote significance between control and indicated test group. For weight loss, significance over the control is shown until maximum difference (**P<0.01, ***P<0.001, ****P<0.0001).

To further characterize these epitopes, the mechanism of antibody protection was determined. Neutralizing capacity was measured by plaque reduction neutralization titers (PRNT$_{50}$). M2e$_{IBV}$ antisera served as a non-neutralizing negative control. HA1$_{231}$ was the only antigen that led to PRNT$_{50}$ values above the limit of detection in all mice (S8A Fig). The NA neutralizing ability of NA$_{328}$ antisera was also assayed (S8B Fig). NA$_{222}$ antisera served as a

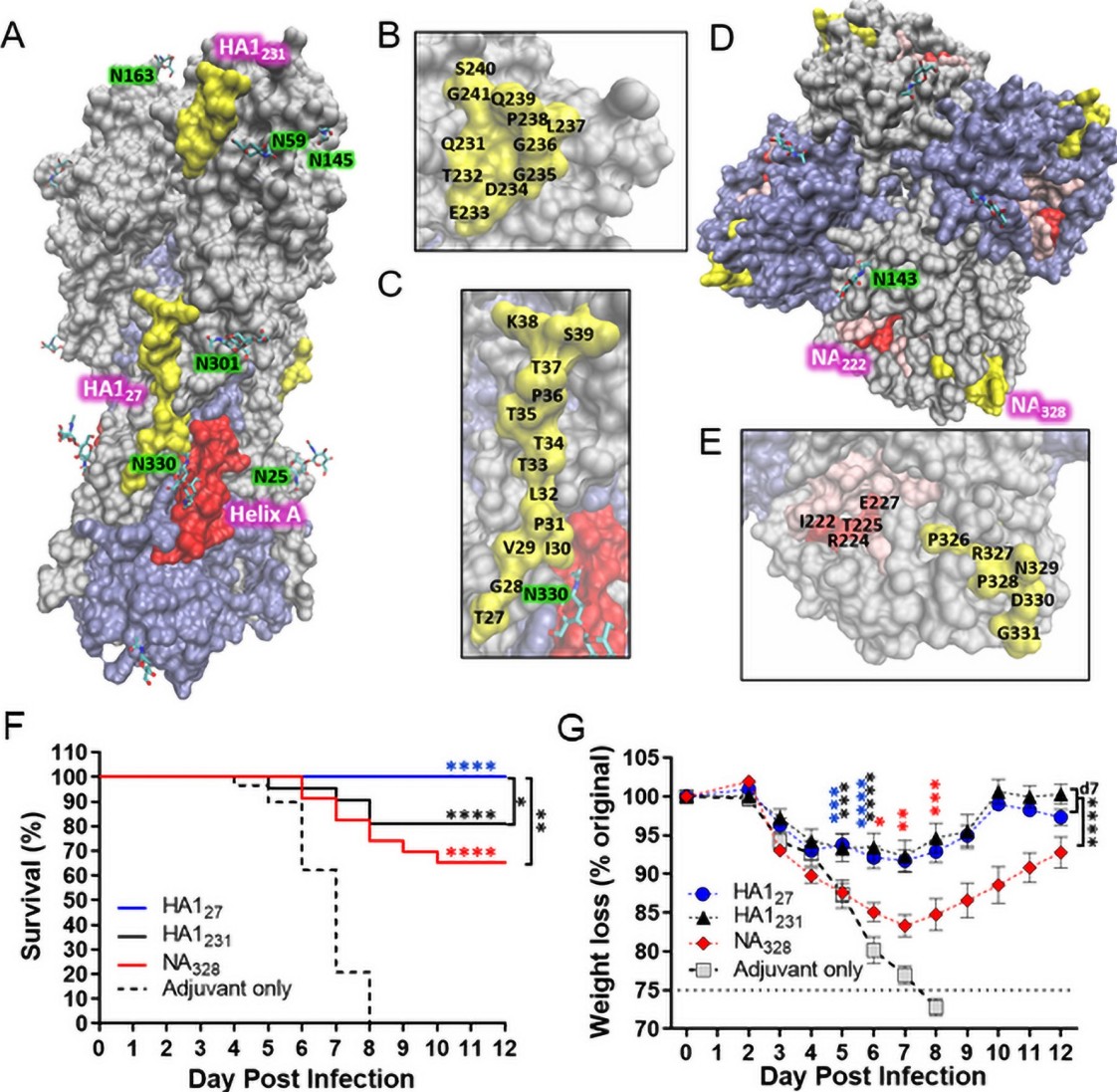

**Fig 3. Identification of 3 pan-IBV antibody epitopes that mediate robust antiviral protection.** X-ray diffraction images of an (*A*) HA trimer and (*D*) NA tetramer. Magnified views of the (*B*) HA1$_{231}$, (*C*) HA1$_{27}$, and (*E*) NA$_{222}$/NA$_{328}$ epitopes are shown, with amino acid residues labeled. Protein subunits are colored grey and purple. Known and putative antibody epitopes are depicted in red and yellow, respectively, and the NA active site is shown in pink. Glycosylated Asn residues are depicted with green highlighting. CD-1 mice were immunized with peptides containing the indicated epitopes plus GLA-SE (or GLA-SE only) and antibody responses were confirmed using d35 sera. On day 42, mice were challenged with 5LD$_{50}$ B/Florida/04/2006 and monitored for (*F*) survival and (*G*) weight loss plotted as mean ± S.E.M. Challenge data is compiled from three experiments (n = 5-8/experiment). Survival curves were compared by log-rank Mantel-Cox test. Data from each weight loss time point were compared by one-way ANOVA followed by Dunnett's multiple comparisons test. Color coded asterisks without brackets denote significance between control and indicated test group; brackets indicate comparison between test groups. Significance between test and control weights are shown for each time point until the group's maximum statistical significance. Weight loss comparisons using brackets represent the most significant difference between indicated test groups, which occurred on the designated day (*P<0.05, **P<0.01, ***P<0.001, and ****P<0.0001).

positive control since this epitope lies in the active site. Each target produced neutralizing antibodies with similar potency against strains representing both Yamagata (B/Florida) and Victoria (B/Malaysia) lineages. These data confirm that the epitope discovery method has antiviral utility and help explain the protection conferred by these vaccines.

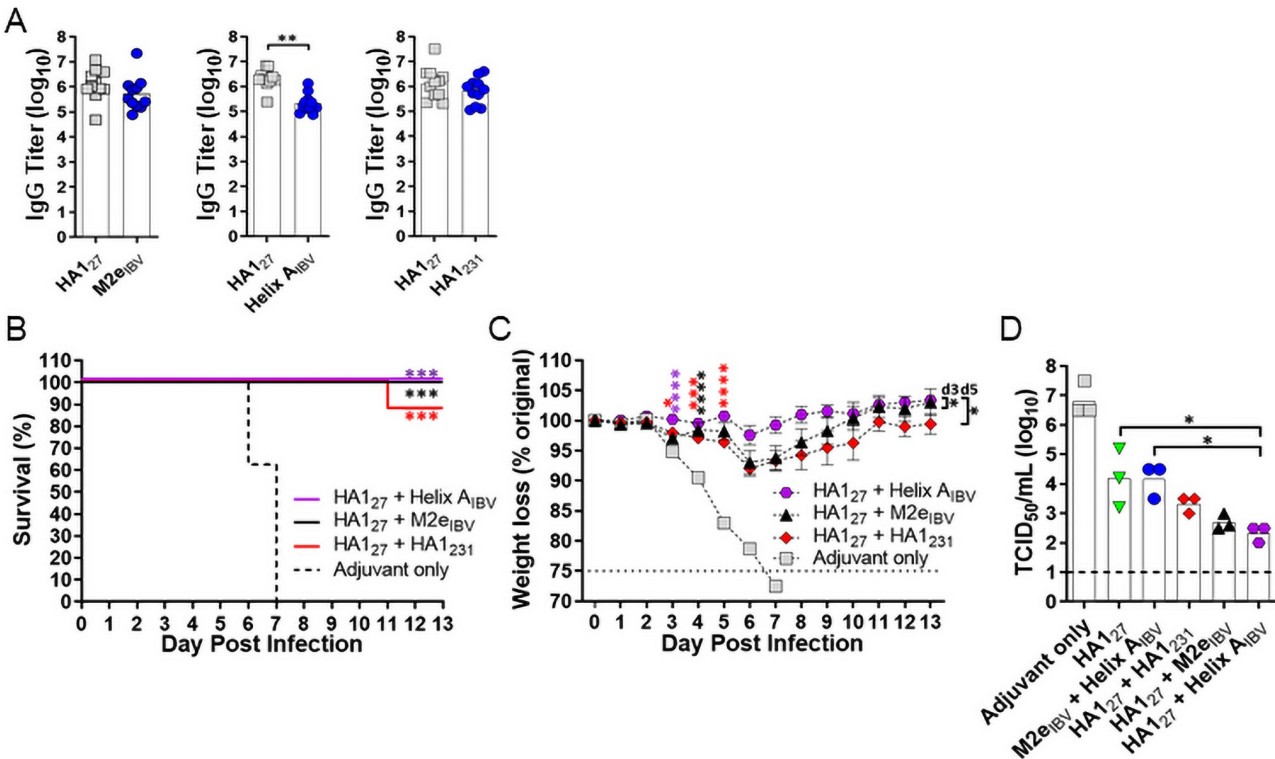

**Fig 4. Bivalent HA1$_{27}$-based influenza B vaccines confer strong protection against influenza B challenge.** CD-1 mice (n = 11) were immunized with HA1$_{27}$ + M2e$_{IAV}$, HA1$_{27}$ + Helix A$_{IBV}$, and HA1$_{27}$ + HA1$_{231}$ formulations (10 µg/peptide) in GLA-SE. (*A*), Day 35 antisera were assayed for titers to each target separately by ELISA. One-way ANOVA followed by Tukey's multiple comparisons test was used for statistical analysis (**$P<0.01$). On day 42, mice were challenged with 5LD$_{50}$ B/Florida/04/2006 and were monitored for (*B*) survival, (*C*) weight loss plotted as mean ± S.E.M, and (*D*) viral load. Survival curves were compared by log-rank Mantel-Cox test. Data from each weight loss time point were compared by one-way ANOVA followed by Dunnett's multiple comparisons test. Color coded asterisks without brackets denote significance between control and indicated test group. Brackets indicate comparison between test groups. For weight loss, significance over the control is shown until maximum difference and comparison between test groups was maximum on the designated day (*$P<0.05$, ***$P<0.001$, ****$P<0.0001$). For viral load, lungs were assayed four days after infection for mean tissue culture infectious dose (TCID$_{50}$), with the limit of detection depicted with dashed line. One-way ANOVA followed by Tukey's multiple comparisons test was used for statistical analysis (*$P<0.05$).

## Construction of maximally protective influenza B vaccines

Given its strong protection, we tested whether HA1$_{27}$ could be paired with the next-best IBV antigens (HA1$_{231}$, M2e$_{IBV}$, Helix A$_{IBV}$) to enhance protection relative to the M2e$_{IBV}$ + Helix A$_{IBV}$ vaccine. Bivalent mixtures exhibited 20–40 nm nanoparticles and stimulated antibody titers to each component (S9 Fig and Fig 4A). Following challenge, the HA1$_{27}$ + Helix A$_{IBV}$ and HA1$_{27}$ + M2e$_{IBV}$ combinations conferred complete survival (Fig 4B). The HA1$_{27}$ + Helix A$_{IBV}$ group showed the least weight loss (Fig 4C) relative to the adjuvant only control and statistical improvement over the other HA1$_{27}$ formulations; in this group, only one mouse lost >5% body weight and all mice showed ≥100% original body weight by day 12. Pulmonary viral loads (Fig 4D) further differentiated efficacy, using the HA1$_{27}$ and M2e$_{IBV}$ + Helix A$_{IBV}$ formulations as positive control references. The HA1$_{27}$ + Helix A$_{IBV}$ combination led to a >10$^4$-fold reduction compared to the adjuvant only group and was the only combination to perform statistically better than HA1$_{27}$ and M2e$_{IBV}$ + Helix A$_{IBV}$ benchmarks. Although it is not clear how broadly applicable this multivalent strategy is for building influenza vaccines, this experiment provides further evidence that targeting two HA epitopes simultaneously can significantly improve antiviral protection.

## Discussion

More broadly protective influenza vaccines would greatly reduce the global health burden caused by seasonal and pandemic outbreaks. Here, we describe a platform for building these vaccines using a ~70 amino acid peptide containing a self-assembling domain, the targeted conserved B cell epitope, and two universal CD4 T cell epitopes that bind a broad repertoire of MHCII molecules [34, 35]. Advantages of this technology include the elimination of non-relevant immunogenic sequences common to conjugate vaccines and VLPs that induce competing antibody responses including carrier suppression [32, 34, 58, 59]. Its manufacture by solid phase peptide synthesis prevents the need for chemical conjugation reactions that require extensive downstream purification [47, 60, 61], and unlike many peptide-based vaccines, it assembles into nanoparticles that facilitate humoral immunity [35, 62].

Here, we measured the protection induced by conserved epitopes on all three influenza surface proteins. Vaccines targeting M2e induced high antibody titers and the two different $M2e_{IAV}$ and $M2e_{IBV}$ sequences mediated equivalent survival (~75%) in their respective challenge experiments. This internal consistency implies that protection was dependent upon its lower copy number relative to HA and NA (~1:60:5 M2:HA:NA) [63] and/or epitope accessibility [64, 65], but independent of challenge strain or epitope sequence. With respect to the two Helix A epitope sequences, Helix $A_{H1}$ induced a high uniform antibody titer, whereas Helix $A_{IBV}$ antibody levels were more variable which may be due to subtle differences in conformation of the IBV epitope or the glycosylation at the nearby Asn residue (N330; see Fig 3), which is not present in H1 HAs [66]. Despite these difference in titers, both epitopes induced the same level of protection that closely approximated the M2e epitopes. The $NA_{222}$ epitope also induced a robust antibody titer, but protection was more limited (~50%) than the M2e and Helix A epitopes following IAV and IBV challenge. Again, this may be due to antibody accessibility since $NA_{222}$ is buried in the NA active site (Fig 3E). Finally, we tested whether protection rates could be improved with bivalent formulations using the M2e and Helix A epitopes, and in both IAV and IBV challenge experiments, overall survival improved to 90% and body weights were better maintained than the monovalent vaccines. Thus, establishing the ability to co-formulate and target 2 epitopes to improve overall survival.

Having validated the technology using known epitopes, we searched for new highly conserved sequences that were readily exposed on the surface IBV hemagglutinin and neuraminidase, which has considerably less sequence variability than IAV [67]. Three sequences, $HA1_{27}$, $NA_{328}$, $HA_{231}$, were selected that are >99% identical across 3,000 independently isolated IBV strains and their induced survival following viral challenge was, respectively, 100%, 80%, and 66%. To the best of our knowledge, this is the first reported characterization of these epitopes, although residues in $NA_{328}$ may reside within the epitope of a recently described anti-H5N1 monoclonal antibody [68]. Interestingly, while these three antisera recognized their peptide-conjugate ELISA reagents equivalently, the antibodies induced by the linear epitope ($HA_{27}$) demonstrated better native protein binding and antiviral protection than the looped/curved epitopes ($HA_{231}$ and $NA_{328}$), thus suggesting a bias in the ability to induce antibodies to linear versus constrained epitopes. In this same experiment $HA_{231}$ and $NA_{328}$ induced similar recombinant protein antibody titers, although $HA_{231}$ stimulated better protection, which could be due to the relative abundance of these two proteins on the virus. Preliminary mechanism of action studies indicated that antibodies directed against $HA_{231}$ neutralized virus infectivity, which may be related to its location near the receptor binding site, and anti-$NA_{222}$ and $NA_{328}$ antibodies inhibited neuraminidase activity. Presumably, ADCC is also an important mechanism of protection given its role in mediating anti-M2e and anti-HA stem antibody activity [48–50]. Futures studies will confirm this and

test whether the novel HA stem binding antibodies directed against $HA_{27}$ and Helix A can also inhibit endosomal fusion [6, 7, 10, 26].

The HA stem is the primary target for building broadly protective influenza vaccines, which is also supported by these studies. The $HA1_{27}$ + Helix $A_{IBV}$ IBV vaccine stimulated 100% survival, negligible weight loss and a $10^4$-fold decline in viral titer relative to controls, and outperformed $HA1_{27}$ formulations with either $HA1_{231}$ or $M2e_{IBV}$, two antigens that showed better monovalent activity than Helix $A_{IBV}$. Targeting these highly conserved IBV epitopes may be superior to existing antigen designs that include stem regions with lower homology or are obscured by glycosylation, the HA head or viral envelope [18, 26, 45]. Future experiments will test whether this improved antibody protection involves coordinated Fc receptor engagement and/or neutralizing activity.

Our method for eliciting antibodies to highly conserved sequences represents a new paradigm for building improved influenza vaccines. Given the putative role that these conserved subdominant epitopes play in maintaining viral function, they should be much less susceptible to mutation. To this point, mAbs specific to conserved epitopes in the HA2 stem [13, 69, 70], NA [71, 72], and M2e [73, 74] are very effective in preventing viral escape. However, the development of vaccines using this approach will require an escape mutant analysis and a need to show protection against multiple strains of virus. This is especially true for epitopes that lie in mutation-prone regions, such as $HA1_{231}$. Establishing protection against strains bearing different glycosylation patterns would also corroborate their utility. Additionally, antisera should be screened against host cells or tissues to test for autoreactivity, as reported for a class of stem-specific B cells [6, 26, 75]. The improved efficacy with bivalent formulations establishes the framework for multi-epitope influenza vaccines, which is also supported by studies showing improved vaccine efficacy following antibody induction to multiple proteins [6, 7, 28–31]. It is also akin to combination monoclonal therapies, where targeting disparate sites on cytomegalovirus, rabies, HIV, Zika, and Chikungunya viruses enhanced antiviral activity and prevented viral escape synergistically [76–79]. Confirmation of the safety and efficacy of this vaccine platform for IBV will support its use for targeting highly conserved epitopes in IAV and other viruses.

## Supporting information

**S1 Table. Peptide vaccine designs used in these studies.**
(TIF)

**S2 Table. Targeted IAV and IBV epitopes in M2e, NA, and HA proteins and their corresponding peptide vaccine design.**
(TIF)

**S3 Table. Novel epitopes in IBV HA and NA (see Fig 3) and their corresponding peptide vaccine design.**
(TIF)

**S1 Fig. Mouse experimentation timeline.**
(TIF)

**S2 Fig. Increasing epitope valency improves antibody responses.** (*A*) Amino acid sequences of M2e antigens. The $M2e_{IAV}$ epitope (italics) was synthesized onto the C-terminus of the peptide monomer (1xC-terminus) or grafted onto two lysine sidechains within the self-assembly domain using isopeptide bonds (2xself-assembly domain). CD4 T cell epitopes from Measles and Hepatitis B are shown in bold. (*B*) Immunogenicity of peptides. CD-1 mice (n = 5)

received a prime-boost immunization with GLA-SE (or GLA-SE only) and d35 titers were assayed by ELISA. A one-way analysis of variance (ANOVA) followed by Tukey's multiple comparisons test was used for statistical analysis (***P<0.001).
(TIF)

**S3 Fig. Peptides targeting conserved influenza A and B epitopes assemble into nanoparticles.** Dynamic light scattering was used to verify nanoparticle size of (*A*) M2e$_{IAV}$, (*B*) NA$_{222}$, (*C*) Helix A$_{H1}$, (*D*) M2e$_{IBV}$, and (*E*) Helix A$_{IBV}$.
(TIF)

**S4 Fig. M2e + Helix A peptide mixtures form nanoparticles.** Dynamic light scattering was used to verify nanoparticle size of (*A*) M2e$_{IAV}$ + Helix A$_{H1}$ and (*B*) M2e$_{IBV}$ + Helix A$_{IBV}$ formulations.
(TIF)

**S5 Fig. Evolutionary sequence profiles of new IBV antibody targets.** The amino acid sequence of each epitope is depicted, with the residue letter height proportional to its mutational frequency in aligned HA or NA sequences. Amino acids are colored according to chemical properties: green (hydrophilic), black (hydrophobic), red (acidic), and blue (basic).
(TIF)

**S6 Fig. Peptides targeting putative influenza B epitopes assemble into nanoparticles.** Dynamic light scattering was used to verify nanoparticle size of (*A*) HA1$_{27}$, (*B*) HA1$_{231}$, and (*C*) NA$_{328}$.
(TIF)

**S7 Fig. Peptides targeting putative IBV epitopes stimulate epitope-specific antibodies that bind recombinant protein.** CD-1 mice (n = 8) were immunized the indicated peptide plus GLA-SE (or GLA-SE only). Antisera (d35) from each group was screened for titers to (*A*) BSA-epitope conjugates or (*B*) recombinant HA/NA. One-way ANOVA followed by Tukey's multiple comparisons test was used for statistical analysis of titers (***P<0.001, ****P<0.0001, n.s. not significant).
(TIF)

**S8 Fig. Neutralization capacity varies by influenza B target.** (*A*) Plaque reduction neutralization titers. CD-1 mice (n = 5) were immunized (d0, d21) with the indicated peptide plus GLA-SE (or GLA-SE only). Day 35 antisera was assayed for neutralizing activity in a PRNT assay. One-way ANOVA followed by Dunnett's multiple comparisons test was used for statistical analysis between control and indicated test group (*P = 0.0156, **P = 0.0044, n.s. not significant). Limit of detection depicted with dashed line. (*B*) NA neutralizing ability. CD-1 mice (n = 3) were immunized as above. Day 35 antisera was assayed for its ability to prevent cleavage of an NA substrate (see Materials and methods). One-way ANOVA followed by Tukey's multiple comparisons test was used for statistical analysis. Color coded asterisks without brackets denote significance between control and indicated test group; brackets indicate comparison between test groups (*P<0.05, **P<0.01, n.s. not significant).
(TIF)

**S9 Fig. Bivalent HA1$_{27}$-based formulations exhibit normal nanoparticle sizes.** Dynamic light scattering was used to verify nanoparticle size of (*A*) HA1$_{27}$ + M2e$_{IAV}$, (*B*) HA1$_{27}$ + Helix A$_{H1}$, and (*C*) HA1$_{27}$ + HA1$_{231}$ formulations.
(TIF)

## Acknowledgments

We thank Dr. Leo Poon (University of Hong Kong) for alignment and homology analysis of influenza B HA and NA sequences.

## Author Contributions

**Conceptualization:** David F. Zeigler, Emily Gage, Christopher H. Clegg.

**Data curation:** David F. Zeigler, Emily Gage.

**Formal analysis:** David F. Zeigler, Emily Gage, Christopher H. Clegg.

**Funding acquisition:** Christopher H. Clegg.

**Investigation:** David F. Zeigler, Emily Gage, Christopher H. Clegg.

**Methodology:** Emily Gage.

**Project administration:** Christopher H. Clegg.

**Supervision:** Christopher H. Clegg.

**Validation:** Emily Gage.

**Writing – original draft:** David F. Zeigler, Christopher H. Clegg.

**Writing – review & editing:** David F. Zeigler, Emily Gage, Christopher H. Clegg.

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
