## [Decision Letter · Decision Letter 0]

11 Mar 2021

PONE-D-21-02059

Epitope-targeting platform for broadly protective influenza vaccines

PLOS ONE

Dear Dr. Clegg,

Thank you for submitting your manuscript to PLOS ONE. After careful consideration, we feel that it has merit but does not fully meet PLOS ONE’s publication criteria as it currently stands. Therefore, we invite you to submit a revised version of the manuscript that addresses the points raised during the review process.

During the revision process, please address the points from the reviewers as part of improving the presentation of the data submitted and the description of how these findings fit into the context of the field.  Specifically, please make certain that the claims made are supported by the data presented.

We look forward to receiving your revised manuscript.

Kind regards,

Victor C Huber

Academic Editor

PLOS ONE

Journal Requirements:

2. Please provide the sequences or accession numbers for all synthesized peptide sequences.

3. Please state what method(s) of euthanasia were used for the animals.

4. Our staff editors have determined that your manuscript is likely within the scope of our Call for Papers on Influenza. This editorial initiative is headed by PLOS ONE Guest Editors Dr. Meagan Deming and Dr. Deshayne Fell. The Collection encompasses research on influenza prevention on every level, including in vitro, translational, behavioral, and clinical studies; disease and immunity modelling; as well as new approaches to influenza prevention. Additional information can be found on our announcement page: https://collections.plos.org/call-for-papers/influenza/.

Currently, your manuscript is included in the group of papers being considered for this call. Please note that being considered for the Collection does not require additional peer review beyond the journal’s standard process and will not delay the publication of your manuscript if it is accepted by PLOS ONE. We would greatly appreciate your confirmation that you would like your manuscript to be considered for this Collection by indicating this in your next cover letter.

If you would prefer to remove your manuscript from collection consideration, please specify this in your cover letter.

5. Thank you for stating in your Funding Statement:

'Financial Disclosure Statement

This research was supported in part by the Center for Disease Control (www.cdc.gov) under award number R43IP001108 (CC). The funders had no role in study design, data collection and analysis, decision to publish, or preparation of the manuscript.'

6. Thank you for stating the following in the Competing Interests section:

'Competing interests

All authors are employees of TRIA Bioscience Corp. DZ and CC are co-inventors on a pending patent (Applicant: TRIA Bioscience Corp., “Synthetic carrier compositions for peptide vaccines”, WO/2020/047107) pertaining to peptide composition of matter and methods of use.'

We note that one or more of the authors are employed by a commercial company: TRIA Bioscience Corp.

c. We also note that you have a patent relating to material pertinent to this article.

In your amended statement of Competing Interests please declare this patent (with details including name and number), along with any other relevant declarations relating to employment, consultancy, patents, products in development or modified products etc. Please confirm that this does not alter your adherence to all PLOS ONE policies on sharing data and materials, as detailed online in our guide for authors http://journals.plos.org/plosone/s/competing-interests by including the following statement: "This does not alter our adherence to  PLOS ONE policies on sharing data and materials.” If there are restrictions on sharing of data and/or materials, please state these.

Please note that we cannot proceed with consideration of your article until this information has been declared.

Reviewers' comments:

Reviewer's Responses to Questions

**Comments to the Author**

1. Is the manuscript technically sound, and do the data support the conclusions?

Reviewer #1: Yes

Reviewer #2: Partly

Reviewer #3: Yes

2. Has the statistical analysis been performed appropriately and rigorously? 

Reviewer #1: Yes

Reviewer #2: I Don't Know

Reviewer #3: Yes

3. Have the authors made all data underlying the findings in their manuscript fully available?

Reviewer #1: Yes

Reviewer #2: Yes

Reviewer #3: Yes

4. Is the manuscript presented in an intelligible fashion and written in standard English?

Reviewer #1: Yes

Reviewer #2: Yes

Reviewer #3: Yes

5. Review Comments to the Author

Reviewer #1: Zeigler and colleagues revised their manuscript according to the reviewers’ comments. It seems that the most critical issues have been addressed and the quality of the paper significantly improved. I only have a few comments that can further improve the manuscript:

1. Lanes 113-114 the sentence is incomplete

2. Lane 148. Please indicate the sex and age of CD-1 mice used in the study. In addition, it would be beneficial for the paper if the authors added a figure with mouse study design, so that the procedures were easier to follow.

3. Lanes 151-155. There should be two separate sentences.

4. Lanes 182-183. Please provide a reference for ELISA method or indicate how the endpoint titers were calculated. Were these antibody of IgG isotype?

5. Figures 1 and 3. Please specify that Y axes show IgG antibody titers (if this isotype was assessed)

6. References 24, 44 are incomplete. Please correct.

Reviewer #2: Figures are blurry to the point where many of the labels are illegible. I did my best to try to figure out what is what based upon context.

Many figures are small, with a single panel or just a few panels, with additional figures in the supplement. I would suggest grouping into multipanel figures and moving as much SI data to the main text as allowable, to make it easier for the reader. For example, Figs 1+2 could be grouped, Figs 3+4 (or even Figs 1+2+3+4); Figs 5+6+S7. Some things, like assessing nanoparticle size/integrity makes sense in the SI,

ln 29-30: In silico method: I think this statement should be removed rom the abstract. It makes it sound like a new computational tool was developed for epitope selection, but ultimately seemed to be making educated guesses based upon structural information, which is not new.

Ln 269-272: “This clearly demonstrates the advantage of targeting influenza epitopes simultaneously.” Fig. 4 shows that the bivalent formulation protects mice better than adjuvant only, but it doesn’t show a comparison between the bivalent and monovalent formulations or whether the bi/monovalent difference is statistically significant. Authors should revise this claim or provide evidence.

ln 356-358: “This evidence confirms…” I would suggest softening this statement. In the same figure there are other examples where targeting 2 epitopes did not result in significantly reduced titers relative to HA1(27) alone, and the HA1(27) + Helix AIBV combination is only significant at the p<0.05 level. So while I agree with the authors overall interpretation of their data, I would say it isn’t clear how generally true this will be with bivalent formulations.

Reviewer #3: This manuscript reports the generation, immunogenicity, and efficacy of conserved peptide epitope-targeting nanoparticle platform vaccines against influenza A and B viruses for broad protection. The peptide monomer vaccines with approximately 70 residues containing 4 IKKIEKR heptad repeats fused to T cell epitopes from Measles virus F2 protein (LSEIKGVIVHRLEGV) and Hepatitis B surface antigens (FFLLTRILTIPQSLD) were chemically synthesized and self-assembled into nanoparticles. Multiple peptide vaccine candidates of monovalent and bivalent epitopes were tested for their immunogenicity and efficacy in CD-1 mice after prime boost vaccination (10 ug dose/peptide). The breadth of protection was assessed with H1N1 influenza A virus and B/Florida virus but not with H3N2 influenza A virus, limiting the significance of this study. Improved efficacy was observed with bivalent peptide vaccines. There are some points to be addressed for clarification.

1. It is difficult to follow this manuscript because of insufficient information on the peptide vaccines tested in this study. The S1 table should be extended to include more details in all peptide vaccines tested: amino acid position (either HA Head or stem) and then consistent numbers of start and end in the peptide epitope sequence given (Helix A-H1, Helix A-IBV, etc.)? – Fig 5 or Fig 6 needs to provide information on the flu B peptide sequences for HA1-27, HA1-231, NA328.

2. It is unclear how much is the sequence homology between the conserved peptide vaccines and challenge viruses. Are all peptide sequences 100% conserved in the challenge viruses tested?

3. Fig 6 data on lung viral titers in TCID50 after challenge with influenza B/Florida: HA1-27 +Helix A-IBV, HA1-27 +M2e-IBV, HA1-27 +HA1-231 show corelates with weight changes (Fig 7). However, the M2e-IBV+ Helix A-IBV (~2 or 3% weight loss, Fig 4) and HA1-27 (7 or 8% weight loss) are not showing correlates with those in groups shown in the Fig 7. Please discuss this apparent discrepancy between body weight changes and lung viral titers.

4. The peptides vaccines contain a self-assembling domain (?) and non-flu pan-T cell epitopes, which is an approximately over 60 residues. Did the authors determine IgG antibodies recognizing the non-flu residues in the vaccines? Are the ELISA coating antigens the same as the vaccine peptides (with a self-assembling domain and non-flu pan-T cell epitopes) or peptide epitopes only specific?

5. The S1 Fig data showed that Increasing M2e epitope valency improved antibody responses. It is not clear whether M2e-IAB (or M2e-IBV, other peptides) contains one copy or 2 copies tested in all other figures data presented.

6. S3 Fig, S5 Fig, S8 Fig Nanoparticle Size distributions: Did the authors include soluble peptides samples (measure the sizes) as a negative control?

7. S4 Fig provides informative Evolutionary sequence profiles of new IBV antibody targets. It will be informative if evolutionary sequence profiles are similarly presented for other peptides vaccines (Fig 1) used in this study.

8. Vaccine doses: Does the bivalent peptide vaccine contain 5 ug each to make it 10 ug in total peptide vaccine dose?

9. Please include representative subtitles in all each figure shown to display the data in a reader friendly manner.

6. PLOS authors have the option to publish the peer review history of their article (what does this mean?). If published, this will include your full peer review and any attached files.

Reviewer #1: No

Reviewer #2: No

Reviewer #3: No

---

## [Author Response · Author response to Decision Letter 0]

22 Apr 2021

please refer to Response to reviewers doc

---

## [Decision Letter · Decision Letter 1]

11 May 2021

Epitope-targeting platform for broadly protective influenza vaccines

PONE-D-21-02059R1

Dear Dr. Clegg,

We’re pleased to inform you that your manuscript has been judged scientifically suitable for publication and will be formally accepted for publication once it meets all outstanding technical requirements.

Kind regards,

Victor C Huber

Academic Editor

PLOS ONE

Additional Editor Comments (optional):

Reviewers' comments:

Reviewer's Responses to Questions

**Comments to the Author**

1. If the authors have adequately addressed your comments raised in a previous round of review and you feel that this manuscript is now acceptable for publication, you may indicate that here to bypass the “Comments to the Author” section, enter your conflict of interest statement in the “Confidential to Editor” section, and submit your "Accept" recommendation.

Reviewer #1: All comments have been addressed

Reviewer #2: All comments have been addressed

Reviewer #3: All comments have been addressed

2. Is the manuscript technically sound, and do the data support the conclusions?

Reviewer #1: Yes

Reviewer #2: Yes

Reviewer #3: Yes

3. Has the statistical analysis been performed appropriately and rigorously? 

Reviewer #1: Yes

Reviewer #2: Yes

Reviewer #3: Yes

4. Have the authors made all data underlying the findings in their manuscript fully available?

Reviewer #1: Yes

Reviewer #2: Yes

Reviewer #3: Yes

5. Is the manuscript presented in an intelligible fashion and written in standard English?

Reviewer #1: Yes

Reviewer #2: Yes

Reviewer #3: Yes

6. Review Comments to the Author

Reviewer #1: The authors addressed all the reviewers' comments and the paper can be published in PLoS One journal. Please ensure that the figures are presented in sufficient quality for good reading.

Reviewer #2: All of my comments have been addressed

............................................................

Reviewer #3: This revised manuscript reports the generation, immunogenicity, and efficacy of conserved peptide epitope-targeting nanoparticle platform vaccines against influenza A and B viruses for broad protection. The peptide monomer vaccines with approximately 70 residues containing 4 IKKIEKR heptad repeats fused to T cell epitopes from Measles virus F2 protein (LSEIKGVIVHRLEGV) and Hepatitis B surface antigens (FFLLTRILTIPQSLD) were chemically synthesized and self-assembled into nanoparticles. The revision has significantly improved the quality and clarification of this manuscript

Most review points previously raised have been addressed.

7. PLOS authors have the option to publish the peer review history of their article (what does this mean?). If published, this will include your full peer review and any attached files.

Reviewer #1: No

Reviewer #2: No

Reviewer #3: **Yes: **Sang-Moo Kang

---

## [Editor Report · Acceptance letter]

18 May 2021

PONE-D-21-02059R1 

Epitope-targeting platform for broadly protective influenza vaccines 

Dear Dr. Clegg:

I'm pleased to inform you that your manuscript has been deemed suitable for publication in PLOS ONE. Congratulations! Your manuscript is now with our production department. 

Kind regards, 

on behalf of

Dr. Victor C Huber 

Academic Editor

PLOS ONE